# Functionalized Mesoporous Silica as Doxorubicin Carriers and Cytotoxicity Boosters

**DOI:** 10.3390/nano12111823

**Published:** 2022-05-26

**Authors:** Carmen Racles, Mirela-Fernanda Zaltariov, Dragos Peptanariu, Tudor Vasiliu, Maria Cazacu

**Affiliations:** 1Department of Inorganic Polymers, “Petru Poni” Institute of Macromolecular Chemistry, 41A Aleea Gr. GhicaVoda, 700487 Iasi, Romania; raclesc@icmpp.ro (C.R.); mcazacu@icmpp.ro (M.C.); 2Centre of Advanced Research in Bionanoconjugates and Biopolymers, “Petru Poni” Institute of Macromolecular Chemistry, 41A Aleea Gr. GhicaVoda, 700487 Iasi, Romania; peptanariu.dragos@icmpp.ro (D.P.); vasiliu.tudor@icmpp.ro (T.V.)

**Keywords:** mesoporous silica nanoparticles, thiol groups, glucose-modified silica, doxorubicin, sustained release, enhanced cytotoxicity

## Abstract

Mesoporous silica nanoparticles (MSNs) bearing methyl, thiol or glucose groups were synthesized, and their encapsulation and release behaviors for the anticancer drug Doxorubicin (Dox) were investigated in comparison with nonporous homologous materials. The chemical modification of thiol-functional silica with a double bond glucoside was completed for the first time, by green thiol-ene photoaddition. The MSNs were characterized in terms of structure (FT-IR, Raman), morphology (TEM), porosity (nitrogen sorption–desorption) and Zeta potential measurements. The physical interactions responsible for the Dox encapsulation were investigated by analytic methods and MD simulations, and were correlated with the high loading efficiency of MSNs with thiol and glucose groups. High release at pH 5 was observed in most cases, with thiol-MSN exhibiting 98.25% cumulative release in sustained profile. At pH 7.4, the glucose-MSN showed 75.4% cumulative release, while the methyl-MSN exhibited a sustained release trend. The in vitro cytotoxicity was evaluated on NDHF, MeWo and HeLa cell lines by CellTiter-Glo assay, revealing strong cytotoxic effects in all of the loaded silica at low equivalent Dox concentration and selectivity for cancer cells. Atypical applications of each MSN as intravaginal, topical or oral Dox administration route could be proposed.

## 1. Introduction

Doxorubicin, the “first line” chemotherapeutic drug, is the most effective anticancer agent against more types of metastatic or early cancer (breast, ovarian, lung, lymphomas or leukemias), despite the severe side-effects including cardiotoxicity, mucocutaneous lesions, myelosuppression and other associated digestive effects, etc. The main pharmaceutical marketed form is the injectable one, due to its low oral availability and permeability, stomach hydrolysis and the enzymatic activation of cytochrome P450 reducing the plasma levels of drugs. Alternative formulations were developed, as, for example, liposomal formulations, in order to support the administration of higher doses, reduced toxicity and prolonged blood circulation time by the efficient evading from the reticuloendothelial system and the enhanced permeation and retention effect, thus ensuring a rapid distribution to the tumor site.

Formulations with reduced side toxicity, increased target delivery, as well as efficiency against multidrug resistant tumors still remain actual challenges, and different solutions were proposed. For example, protein (apo-transferrin and lactoferrin)-conjugated doxorubicin formulations showed efficacy on hepatocarcinoma cells by oral administration [1]. Dox-NPs based on poly(lactic-co-glycolic acid) (PLGA) proved a higher availability (363%) of Dox in oral administration, that could be due to the bypassing of the P-glycoprotein transporters and the favored absorption on microfold cells (M-cells) in the intestine, which are the main cause of the poor permeability of Dox [2]. Cardiotoxicity of Dox can also be reduced by conjugation with lipids. Thus, simvastatin/Dox formulations in HeLa cells and Dox-SLN (solid lipid nanoparticles) for melanoma proved therapeutic efficiency in local administration, increasing the drug’s permeation though the uterine cervix, mucosa and epidermis, respectively [3].

Besides using liposomes, nanoparticles [3,4,5] or porous scaffolds [6], administration routes other than per injection are also taken into account, as, for example, oral delivery [1,2,7], implantation [6], dermal administration [3] and/or combined delivery mechanisms [5]. The targeted release of Dox can be ensured by the implanting of drug carriers at the site of the tumor. Silica particles offer a good alternative to the organic (cellulose, chitosan-based hydrogels) delivery systems. Silica materials are biocompatible with a rate of hydrolysis of 76% in 56 days and urinary excretion and offer an excellent protection of drugs by internal loading, ensuring a prolonged effective concentration [8]. These can be successfully applied as transmucosal drug delivery systems even in a controlled way, being stable in the gastrointestinal medium and promoting a site-specific release of the drug [9].

Ordered mesoporous silica materials are largely used as drug carriers, due to their intrinsic properties, such as chemical stability, high surface area, biocompatibility and large pore volume. In addition, there are several possibilities for adjusting the properties from synthesis or by post-synthesis functionalization, aiming to tune the size and chemistry of the pores, shape, size and functionality of the nanoparticles or to achieve surface modifications for targeted medical applications [10]. Mesoporous silica nanoparticles (MSNs) are considered safe for biological applications [11], but their safety and biodistribution are still under investigation. Their toxicity depends on the size and shape, chemical composition and administration route. It seems that silica nanorods mainly store in the liver and spleen, while the nanospheric particles were found mostly in the spleen. While the subcutaneous, oral and ocular administration proved to be safe, the intraperitoneal and intravenous administration may cause thrombosis [9]. A cytotoxic effect by the apoptosis and necroptosis mechanisms was observed for nano-SiO_2_ on hepatocarcinoma and breast cancer lines [12]. In spite of such concerns, the relative lower cytotoxicity of mesoporous silica, the high cellular uptake by endocytosis and the ability to increase their functionalities by thermo-, pH- and redox-responsive gating groups made them valuable candidates for drug delivery systems [13].

So far, MSNs were designed as targeted drug carriers for the release of drugs into cancer cells, overcoming the MDR (multidrug resistance) effect. Encapsulation of the doxorubicin (Dox) into mesoporous silica was extensively studied [14]. The in vivo biodistribution showed that such formulations induced a higher accumulation of Dox in drug resistant tumors than free Dox [14]. However, the incomplete delivery of the drugs over time can induce bioaccumulation, promoting toxicity and limiting the therapeutic effects.

Monodisperse mesoporous silica were used as a coating for superparamagnetic iron oxide nanoparticles, showing increased efficiency as a contrast agent for magnetic resonance imaging (MRI), partial permeability to water molecules and no toxicity to cells [15].

Further conjugation with transferrin (Tf) of the magnetic Fe_3_O_4_ NPs embedded in Pluronic F127 revealed a temperature-dependent release of Dox [16]. Photosensitizer-conjugated MSNs were proposed for use in combining chemo- and photodynamic therapies. The cytotoxicity evaluated by the ROS level of the conjugated particles in cancer cells (A549 lung cancer) showed an increase of about 17 times upon irradiation (λ = 660 nm, 2 min) [17].

Surface conjugation of silica by the attachment of adamantylamine involving disulfide bridges and post-functionalization with cyclodextrin-PEG shell structures was found to be a good strategy for loading (LC 79%), and redox triggered the prolonged release of Dox up to 120 h in PBS 7.4, even if the toxicity of Dox severely decreased by encapsulation [18]. Thermo- and pH-responsive functionalized core-shell silica NPs based on responsive poly(N-isopropylacrylamide) (PNiPAm)-co-acrylic acid (AA) hydrogels showed an improved Dox delivery by increasing the acidity and temperature [19].

It is known that thiol groups can bind to proteins and influence their stability and solubility [20]. Thiol-modified silicones showed increased muco-adhesion, based on their bonding abilities via disulfide linkages [21], which would promote the increase of the drug residence time. β-Cyclodextrin modified MSNs with amine, hydroxyl and thiol groups wereinvestigated for potential use in intravesical chemotherapy, and a sustained release of the anticancer drug was observed when the thiol groups were present, promoted by the formation of the disulfide bonds with the bladder mucosa [22]. Gold NPs-functionalized mesoporous silica with thiol groups exhibited a high Dox encapsulation and a pH-responsive release at pH 5.5, corresponding to the intracellular environment of cancer cells, upon irradiation by NIR laser (λ = 808 nm) [23].

The efficiency of Dox in tumor cells can often be limited, mostly by drug resistance phenomenon. Glucose metabolism in these cells was found to be higher, so that a reduction in glucose concentration would significantly reduce the cell viability. A glucose dependence of the cell viability in the presence of Dox was reported for chondrosarcoma cells, where the glucose uptake had a positive impact on the resistant cells. The main mechanism involved in the drug resistance was highlighted by the overexpression of some transmembrane proteins: P-glycoprotein (involved in the efflux of the drugs from cells); HuR (an RNA binding protein involved in post-transcriptional regulation of mRNAs) and GLUT (glucose transporter) proteins’ family involved in glucose/different carbohydrate metabolism, ensuring homeostasis of the body [24]. The conjugation of glucose with chemotherapeutic drugs can be used as a strategy to improve cytotoxicity in cancer cells. Thus, Dox-loaded poly(amidoamine) dendrimer modified with glucose groups showed an increased cytotoxicity in glucose-deprived MCF-7 cells, due to the Warburg effect and specific tumor accumulation, while overexpression of GLUT proteins in adenocarcinoma and glioma cancer cells was found to influence the endocytosis of Dox-loaded glucose-conjugated NPs, targeting the tumor and the blood–brain barrier [25].

Since protein–carbohydrate interactions are mediated via lectin receptors, which are overexpressed on the surface of neoplastic cells, this interaction could be used in targeted chemotherapy, as was reported for gelatin mannosylated-Dox-mesospheres [26]. It was also observed that the interaction between Dox and hemoglobin is influenced by the presence and concentration of glucose [27] reducing the binding affinity of Dox to hemoglobin by allosteric effect, and suppressing the impact of Dox on hemoglobin structure.

Given the role of thiol and carbohydrates in drug delivery systems, in this study we prepared functionalized mesoporous silica particles with thiol and glucose groups, by unsophisticated and green synthesis method. These, as well as a mesoporous silica with methyl groups, can be proposed as highly efficient carriers for the loading and release of Dox, prompted exclusively by physical interactions, which were simulated and measured experimentally. The structural features, the morphology of pores and specific BET area, the surface charge and the loading and release profiles of Dox were evaluated by specific characterization methods. The drug release behavior was evaluated at different pH values (2.6, 5 and 7.4) mimicking the physiological medium in specific organs/tissues. The in vitro cytotoxicity of the free and loaded silica was determined by CellTiter-Glo^®^ assay on normal and cancer cell lines (NHDF, MeWo and HeLa) to test the carrier efficiency in specific local applications.

## 2. Materials and Methods

Cetyltrimethylammonium bromide (CTAB), tetraethoxysilane (TEOS), methyltriethoxysilane, 3-mercaptopropyltrimethoxysilane, D-glucose (mixture of anomers), 2-allyloxyethanol (AE), 3-mercaptopropyl methyldimethoxysilane (R-SH), Amberlite IR-120(plus)—a strongly acidic gel-type resin with sulfonic acid functionality, having total exchange capacity of 1.9 meq/mL—from Sigma-Aldrich and 2,2-dimethoxy-2-phenylacetophenone (DMPA) from Merck were used as received. Doxorubicin hydrochloride concentrate 2 mg/mL for solution infusion was purchased from Accord Healthcare. Phosphate Buffers (PBS) of different pH were prepared from the following ingredients, which were high purity compounds supplied by Aldrich: PBS of pH 7.4: NaCl (8 g), KCl (0.2 g), Na_2_HPO_4_ (1.44 g) and KH_2_PO_4_ (0.24 g) in distilled water (1 L); PBS of pH 5:citric acid monohydrate (9.605 g) and Na_2_HPO_4_ (18.15 g) in distilled water (1 L) and PBS of pH 2.6: citric acid monohydrate (18.238 g) and Na_2_HPO_4_ (1.875 g) in distilled water (1 L).

Nitrogen adsorption–desorption isotherms of functionalized silica were recorded with an AutosorbiQ Station1 device from QuantaChrome Instruments.

Transmission electron microscopy (TEM) images were obtained with a Hitachi HT7700 microscope, operated in high contrast mode at 100 kV accelerating voltage.

^1^H and ^13^C-NMR measurements were executedin D_2_O at room temperature on a Bruker NEO-1 400 MHz spectrometer, equipped with a 5 mm four nuclei (^1^H/^13^C/^19^F/^29^Si) direct detection probe.

Fourier Transform Infrared (FT-IR) spectra were registered on a Bruker Vertex 70 Spectrometer in transmittance mode, in the 400–4000 cm^−1^ spectral range, with a resolution of 4 cm^−1^ and accumulation of 32 scans. Spectra processing consisting of atmospheric compensation, baseline correction, normalization, and 2nd derivative, was made by OPUS 6.5 software.

Raman spectra were registered by using an inVia confocal Raman microscope (Renishaw Apply Innovation) in the 180–3200 cm^−1^ spectral range, with Raman scattering excitation wavelengths 633 nm and 785 nm diode laser systems.

TGA-DTG analysis was executed in nitrogen, on a Discovery TGA 5500 analyzer from TA Instruments, with a 10 °C/min ramp.

Molecular dynamic simulations of the interaction between the functional groups and Dox were performed using the YASARA-Structure software package version 19.9.7 (YASARA-Biosciences GmbH, Vienna, Austria) [28] that comprised the “AutoSMILES” algorithm for automatic parameterization of the studied compounds. Hence, this algorithm was used to generate the force field parameters for the molecular dynamic simulations using the YASARA force field. The structures of two molecules mimicking functional fragments in silica and of Dox were built using the YASARA builder. Each fragment was then added to the simulation box in the presence of a Dox molecule, with a distance of 20 Å between them to eliminate any biased interactions at the start of the simulation. The simulations were done in vacuum, to better highlight the interactions between the compounds, at a constant temperature of 298 K. The simulations had a length of 120 ns and the final conformation was visually analyzed using the VMD software [29].

Fluorescence spectra were registered in 10 mm quartz cuvettes in solution (PBS pH 7.4) on a Duetta™ spectrometer, equipped with CCD detector and a 75 mW Xenon light source.

UV-Vis spectra of Dox solutions before and during the release studies were measured in PBS pH 2.6, 5 and 7.4, in 10 mm quartz cuvettes on a Specord 210 Plus spectrophotometer (Analytic Jena).

The Zeta potential of the silica before and after encapsulation of Dox was measured on a Malvern Zeta-Sizer IV equipment (Malvern Instruments) at room temperature, in distilled water.

Statistical analysis of the Dox release at different pH values (2.6, 5 and 7.4) was used to investigate the differences among means. A statistical tool based on variance analysis (ANOVA) was applied to establish and measure the effects of various experimental factors on the obtained results of the research. ANOVA analysis was performed for a 5% confidence interval, the “α” parameter being set to 0.05. If the *p*-value (probability that the null hypothesis is true) is less than 0.05, the null hypothesis is false, meaning that such interactions (with *p* ≤ 0.05) show a synergy between the independent factors, revealing that the results are replicable. When the *p*-value is higher than 0.05, no effect was observed and the null hypothesis is confirmed.


**Cell culture**


Normal human dermal fibroblasts (NHDF) were purchased from PromoCell; MEWO and HeLa from CLS Cell Lines Service GmbH (Eppelheim, Germany); Eagle′s Minimal Essential Alpha Medium (aMEM), Dulbecco′s Modified Eagle Medium (DMEM) without red phenol, 1% Penicillin-Streptomycin-Amphotericin B (10K/10K/25 µg in 100 mL) from Lonza; fetal bovine serum (FBS) from Biochrom GmbH, Germany; cell proliferation and cytotoxicity test CellTiter-Glo^®^ from Promega; Tryple from Thermo Fisher Scientific;phosphate buffered saline (PBS) from Invitrogen; Human Serum Albumin >98% from Aldrich.

Cells were cultured in a complete medium consisting of alpha-MEM, which was supplemented with 10% FBS and 1% mixture of penicillin-streptomycin-amphotericin B in culture vessels with the surface treated for tissue cultures, under humidified atmosphere with 5% CO_2_ at 37 °C until a sufficient number of cells were obtained for the experiments. For passage, the cells were washed with PBS and detached with Tryple, resuspended in complete medium and centrifuged at 200 g, 3 min, after which they were recultivated in new vessels and a complete medium.


**CellTiter-Glo^®^ assay**


Prior to the CellTiter-Glo^®^ test, cells were seeded on 96-well opaque white plates at a concentration of 5 × 10^3^ cells/well for the NHDF line and 10 × 10^3^ cells/well for the HeLa and MeWo cell lines, after which they were incubated for 24 h in atmosphere humidified with 5% CO_2_ at 37 °C. The next day, the medium was replaced with Dox, silica (MD1, Sil2, Sil3, MSil2, MSil3) and silica-Dox (MD1D, Sil2D, Sil3D, MSil2D, MSil3D) solutions. Control samples were treated with complete medium only. The plates were incubated for 48 h in an incubator at 37 °C and 5% CO_2_. At the end of the 48 h, the plates were removed from the incubator and equilibrated at room temperature for 30 min, after which 100 µL/CellTiter-Glo^®^ reagent was added and the plates were re-incubated at room temperature for another 15 min. After this last incubation, the plates were inserted into the plate reader (FLUOstar^®^ Omega, BMG-Labtech) and the intensity of the emitted light was read. The relative viability of the cells was estimated as the percentage of RLUs (relative light units) of the samples in relation to the RLUs of the control wells, according to the formula:

Relative viability (%) = 100 × (RLU_p_ − RLU_blk_)/(RLU_c_ − RLU_blk_), where RLU_p_, RLU_c_, RLU_blk_, represent the RLUs of the samples, control and blank, respectively. The experiments were performed in duplicate and repeated three times and the data analysis was performed with GraphPad Prism software version 8.00 for Windows (GraphPad Software, San Diego, CA). The CellTiter-Glo^®^assay procedure involves the addition of the single reagent (CellTiter-Glo^®^ Reagent) directly to the cells grown in the medium. The addition of the CellTiter-Glo^®^ reagent results in cell lysis, and the generation of a luminescent signal proportional to the amount of ATP present, while the amount of ATP is directly proportional to the number of cells present in the culture.

### 2.1. Synthesis of Mesoporous Silica

Two mesoporous silica materials: MSil1 and MSil2, with methyl and 3-mercaptopropyl organic groups, respectively, attached to silicon, were obtained, following in general terms a previously reported procedure [30], that is, co-hydrolysis of alkoxysilanes using CTAB surfactant as template, in alkaline medium, followed by template elimination in acidic conditions.

Allyloxyethyl-glucopyranoside derivative (AG) was synthesized, following the same method as described before [31], i.e., by Fischer glycosylation with allyloxyethanol, using Amberlite IR-120 Plus cation-exchange resin as the catalyst. The^1^H-NMR (D_2_O, *δ* ppm): 5.92–6.02 (m, 1, CH–); 5.38–5.27 (m, 2, CH_2_); 4.95, 4.52 (d, 1, H anomeric, two isomers); 3.4–4 (m, 12, –CH_2_ –, –CH*<*); ^13^C-NMR (D_2_O, *δ* ppm): 133.5 (=CH–); 118.5 (=CH_2_); 102.3, 98.3 (C^1^ anomeric, two isomers), 75.8 (C^4^); 73.0 (C^3^); 71.7 (C^2^); 69.6 (CH_2_–CH=); 68.9 (CH_2_–CH_2_–O); 66.7 (C^5^); 62.5 (O–CH_2_–CH_2_); 60.5 (C^6^).

The synthesis of MSil3 was executed by thiol-ene photoaddition, as follows: to a solution of AG (260 mg in 16 mL distilled water), 84 mg of MSil2 and 9 mg of DMPA were added and the mixture was UV-irradiated at 365 nm for three periods of 10 min each, intercalated with 1 min sonication with a UTR200 ultrasonic processor at 0.4 cycle and 40% amplitude. The silica was centrifuged at 6000 rot/min for 10 min, separated, washed with water with centrifugation three times, separated and dried in a vacuum.

### 2.2. Drug Encapsulation

Dox loaded silica NPs were prepared by using 10 mg of silica NPs immersed in 5 mL of Dox solution 0.1% in PBS 7.4. The silica was stirred in a rotator shaker (25 °C, 150 rpm) in light-sealed conditions for 24 h. The loaded silica was then centrifuged, washed with 2 mL of distilled water and dried in the dark. The loading capacity (LC) and encapsulation efficiency (EE) were calculated by using the Dox absorbance wavelength at 482 nm, according to the Dox calibration curves at pH 7.4 (ε = 5498 M^−1^ cm^−1^). In comparative experiments, the encapsulation was completed from a more concentrated Dox solution, that is ~0.2%, and the mesoporous silica MSil2 and MSil3, which reached 100% EE, were submitted to a second encapsulation.

The encapsulation efficiency (EE) and silica loading capacity (LC) were calculated according to Equations (1) and (2), respectively:EE (%) = 100 × (C_0_ − C)/C_0_(1)
LC (µg/mg) = (C_0_ − C) × 543.5 × V × 10 ^3^/M_e_(2)
where: C_0_ = initial amount of Dox (M); C = amount of Dox in supernatant after encapsulation (M); 543.5 = molar mass of Dox; V = volume of Dox solution used (mL) (typically V = 5); M_e_ = amount of silica used for encapsulation (mg) (typically M_e_ = 10)

### 2.3. In Vitro Drug Release Study

The loaded silica NPs (7 mg) were dispersed into 2 mL PBS (at different pH values: 2.6, 5 and 7.4) and stirred at 37 °C under dark conditions. At certain intervals, the suspension was centrifuged and the release medium was collected and replaced by an equivalent volume of fresh PBS medium. The concentration of the released Dox was determined by UV-Vis measurements (absorption band at around 490 nm) based on the specific calibration curves at each pH value. Thus, the evaluation of the release behavior was completed using, in each case, the extinction coefficient measured at the corresponding pH, i.e.,: ε = 5498 M^−1^ cm^−1^ for pH 7.4, 8361 M^−1^ cm^−1^ for pH 5, and 9894 M^−1^ cm^−1^for pH 2.6, respectively. For the comparative experiments with silica loaded with 0.2% Dox, the release was followed at pH 5.

The cumulative release, and the amounts of drug released reported to the mass of silica were calculated with Equations (3) and (4):Cumulative release (%) = 100 × C_r_ /(C_0_ − C)(3)
Amount released (µg /mg silica) = LC_r_ × C_r_/(C_0_ − C)(4)
where: C_r_ = concentration of Dox released (M); C_0_ = initial amount of Dox (M); C = amount of Dox in supernatant after encapsulation (M); LC_r_ = loading degree calculated for the silica amount used in release experiment.

The drug release kinetics were analyzed with two mathematical models, Higuchi (Equation (5)) [32] and Korsmeyer–Peppas (Equation (6)) [33].
M_t_ = K_H_ t^1^/^2^(5)
M_t_/M_∞_ = K_KP_t^n^(6)
where M_t_ is the amount of drug released at time t; M_∞_ is the initial amount of drug in the sample; K is the rate constant; and n is the release exponent, which characterizes the drug release mechanism [34].

## 3. Results and Discussion

### 3.1. Synthesis and Characterization of Mesoporous Silica

Five silica materials were prepared, adapting a procedure from the literature [30], followed by chemical modification (Figure 1), that is, a silica with ca. 10% mercaptopropyl groups, (Sil2), a mesoporous silica derived from this one (MSil2), two silica carrying glucoside-modified functional groups, derived from Sil2 and MSil2, coded Sil3 and MSil3, respectively, and a mesoporous silica with ca. 27% CH_3_ groups (MSil1). The first step in their synthesis was the co-condensation of tetraethoxysilane with 3-mercaptopropyltrimethoxysilane or methyltrimethoxysilane in the presence of CTAB, which afforded the nonporous silica, (Sil2, Sil3), followed in the second step by template extraction to generate the mesopores. The chemical modification of the SH groups with allyloxyethylene-glucopyranoside [31] was performed by thiol-ene addition, combined with ultrasound treatment, in water, in the presence of 2,2-dimethoxy-2-phenylacetophenone (DMPA) as photoinitiator.

In this contribution, we focused on mesoporous silica materials, but used the nonporous ones (Sil2 and Sil3) as comparison terms. The main characteristics of the mesoporous silicas are presented in Table 1, as determined by the nitrogen sorption–desorption isotherms and TEM.

As can be observed from the TEM images in Table 1, the methyl-functionalized silica presented quasi-spherical nanoparticles, in the range of 100 nm in diameter, while de thiol-functionalized silica showed spherical and elongated (rod-like) particles having ca. 100 nm in diameter and ca. 1 μm in length. After modification with glucose, the particles were slightly larger, with more pronounced aggregation tendency. The rounded hexagonal section can be noticed in both MSil2 and MSil3. Based on the sorption–desorption curves (Appendix A), the type of the isotherms and the shape of the pores were assigned, according to IUPAC classification, as type IV with very small hysteresis loops indicating slit pores [35]. The BET analysis data indicate pore average at the limit of the microporous and mesoporous materials (around 2 nm).

To confirm the addition reaction between allyloxyethylene-glucopyranoside (AG) and mercaptopropylsilane moieties, a similar reaction was performed using 3-mercaptopropyl methyldimethoxysilane (R-SH) instead of (M)Sil2, which gave a soluble compound. Its ^1^H-NMR spectrum showed complete disappearance of the double bond and of the SH signals, confirming that the established conditions ensured complete reaction.

In the FT-IR spectrum of the initial SH-containing silica, the absorption bands characteristic to SH groups which appear at 2600–2540 cm^−1^ [36] were very weak and could not be used as a reference for the chemical modification. The broad band at 3450 cm^−1^ is due to adsorbed water and Si-OH groups, while the bands at 2928–2855 cm^−1^ are assigned to asymmetric and symmetric stretches of C-H bonds, more pronounced in MSil3. The main bands at 1052–1080 cm^−1^ and 952–964 cm^−1^, assigned to the Si-O stretches, are characteristic of the silica framework. In the 400–800 cm^−1^ range, small bands assigned to C-S-C stretching vibrations (662 cm^−1^and 694 cm^−1^)appeared (Appendix A), while other bands characteristic for the glucose derivative—besides those of OH and C-O stretching, which overlap with the major silica bands—confirmed the presence of the functional groups (1730, 1690, 758 cm^−1^). The -SH groups in Sil2 and MSil2 and the glucose fragment in Sil3 and MSil3 were highlighted by the absorption maxima obtained on the second derivative of the spectra in the 850–400 cm^−1^ (Appendix A), with absorption maxima at 762 cm^−1^ and 716–700 cm^−1^.

In the Raman spectra of Sil2 and MSil2, the SH band was visible at 2576 cm^−1^ and disappeared in Sil3 and MSil3 (Figure 1). A strong band at 507 cm^−1^ was noticed in MSil2 spectrum, which could be due to the labile disulfide S–S bridges, possibly formed during storage in air [37]. The newly formed thioether bonds were evidenced by the apparition of some bands in the range ~ 600–700 cm^−1^ (C-S in aliphatic sulfides) and at 1213 cm^−1^ (CH_2_-S wag). The bands at 1240–1340 and 1460 cm^−1^ are assigned to the deformations of CH_2_ groups [38] and the bands at 1410–1420 cm^−1^ are specific for Si-C bonds. The Si-O-Si symmetric bending vibrations can be observed at 780–800 cm^−1^, the siloxane ring at 490 cm^−1^ and Si-OH groups at 980 cm^−1^ [38]. The new strong band at ~1000 cm^−1^ and the lower peaks at 1040 cm^−1^ and 1160 cm^−1^ are due to the C-O bonds in the glucoside moiety [39].

As observed in the model reaction and confirmed by the Raman spectra, all of the available SH groups are likely to be modified in the thiol-ene photoaddition, but a certain percent of the initial SH groups were already trapped in disulfide bridges. So, the amount of glucoside derivative chemically linked to the mesoporous silica MSil3 was evaluated by comparing its TGA curve with that of the initial thiol-functionalized mesoporous silica (MSil2). The more pronounced weight loss observed in MSil3 is normal, knowing the weak thermal stability of saccharides and ether groups. We assigned the difference between the residues at 700 °C, representing ~12 wt.%, to the loss (thus the weight content) of the glucose derivative (Figure 2). This percent would represent ca. 3.3 mol% reported to silica’s composition. When the DTG curves are analyzed, a new peak can be observed for MSil3, centered at 250 °C, while the last decomposition peak at around 520 °C in MSil2 is wider and shifted to a lower temperature (~460 °C) in MSil3. Taking into account the structural differences between the two samples, we assigned the former to decomposition of glucose groups, and the latter to cleavage of the aliphatic segment, which in MSil3 is longer, containing supplementary ethyleneoxypropylene moieties.

Even though the level of chemical modification with glucose is relatively low, this could imply significant modification concerning the hydrophilicity of the silica, while the glucose moieties might have an important role in the improvement of the water solubility and the stability of the drugs in the biological environment, possibly easing the docking of the doxorubicin conjugate to the cancer cells, based on their increased affinity for glucose, or contributing to pH modification in solid tumors [40]. According to the structure, the methyl groups in MSil1 are the most hydrophobic organic groups, while glucose is a strongly hydrophilic molecule. Indeed, significant difference of the wetting behavior was noticed between the samples just after the immersion in doxorubicin solution, or when the water drops were cast on the powders (Appendix A). The most hydrophobic sample was MSil1, which remained on the surface of the solution, while the most hydrophilic one was MSil3, containing glucose, which was perfectly dispersed. A better wetting was visually observed for the mesoporous silicas, compared with their nonporous counterparts, in both series. An estimation of the water contact angles showed values of around 94° for MSil1, 80° for MSil2 and 67° for MSil3. After a while, all of the silica got wet and well dispersed in the Dox solution.

### 3.2. Interactions between Dox and Functional Silica

The encapsulation of Dox within all of the silica materials is due exclusively to physical interactions, more or less pronounced, depending on chemical structures: hydrogen bonding; electrostatic interactions and hydrophobic interactions. Based on the acidity constants of the main components of the system (Table 2), one can estimate the nature of these interactions. At pH 7.4, as used in our encapsulation experiments, the amine groups from Dox are mostly protonated, the Si-OH groups are partly ionized [41], while the OH groups in glucose and the SH groups in mercaptopropyl are protonated as well. It follows that in all of the silica materials Dox is bound through the electrostatic interactions of the positively charged protonated primary amino groups with the negatively charged silanol groups in the pores, as well as by the hydrogen bonds between the OH groups in Dox and oxygen from the siloxane framework, and between the protonated silanol and carbonyl groups in Dox [42]. In addition to the relatively similar interactions with the silica framework in all of the cases, Dox may present supplementary physical bonding with the thiol-and glucose-functionalized silica.

### 3.3. Molecular Dynamics Simulations

In order to determine how the functional groups, present in MSil3 and MSil2, influence the interaction with Dox, we performed molecular dynamics simulations (MD), using the YASARA software. To this end, two organic fragments containing the functional group specific to each material were used in the MD simulations to understand the interaction between Dox and modified silica. The simulations were performed in vacuum, to better emulate the “shelf conditions”, and had a length of 120ns. Figure 3 depicts the conformations obtained at the end of the MD simulations. It can be seen that, in the case of the thiol-ending fragment, the dominant interaction is of a hydrophobic nature, between the alkane tail of the functional fragment and the aromatic rings of Dox. The fragment is also able to make a hydrogen bond between the -SH group and an oxygen atom in Dox (Figure 3A, green dotted line). In the case of the glucose modified fragment, a different behavior can be seen. The hydrophobic interaction is replaced by many hydrogen bonds that form between the -OH groups of glucose and the O atoms in Dox (Figure 3B, red dotted lines). In both of the types of modified silica materials, it is expected that the functional groups ensure a better encapsulation of Dox. It was postulated that the photostability of Dox increases in silica, due to a high local concentration of the drug in the pores and at the surface of the particles [46,47]. Thus, the enhanced physical interactions could increase the shelf stability of the loaded silica in a dry state.

We also tried to perform the simulations in water, but no stable formation of the complexes between Dox and the organic fragments mimicking functionalized silica was observed. This was due to a competition in the hydrogen bond formation between Dox, the organic fragments and the water molecules. This is, however, only a partial image of the mode of interaction between Dox and MSil3 or MSil2. We expect that the silica framework with its un-condensed -OH groups, together with the porosity, would have an important role in the adsorption of Dox, but to take into account all of these parameters is a very challenging task.

### 3.4. Fluorescence

Further evidence for the self-associating behavior of Dox with glucose and thiol precursors in solution was obtained by fluorescence spectroscopy. The absorption spectra in PBS 7.4 (Appendix A) revealed a shoulder maximum at 270 for AG assigned to n→π* transitions of carbonyl group, and maximum at 500 nm in Dox and Dox: AG 1:1 mixture attributed to the π→π* transitions of hydroquinone ring (Appendix A). The excitation spectra of Dox solution 0.1% in PBS 7.4 revealed the presence of two maxima at 270 nm and 470 nm. The emission spectrum of Dox at λ_exc_ = 270 nm consisted of two maxima at 350 nm and 596 nm, with two broad shoulders at 555 nm and 632 nm, while the emission spectrum of AG revealed a maximum at 358 nm at λ_exc_ = 270 nm. The maximum at 596 nm (with its “shoulders”) also appeared at λ_exc_ = 470 nm in the spectrum of Dox, and was due to the dihydroxyanthraquinone fragment (Appendix A). The emission spectrum of Dox: AG 1:1 (Appendix A) at λ_exc_ = 270 nm showed, besides the maxima at 350 nm and 596 nm due to the Dox fragment, two shoulder bands at 496 nm and 505 nm assigned to the interaction of Dox with AG. At λ_exc_ = 470 nm, the emission maximum of Dox at 596 nm was accompanied by a new band at 514 nm. In order to highlight that the maxima at 514 nm was due to the association of Dox with AG, we performed some titration tests of Dox solution 0.1% in PBS pH 7.4 with AG solution 0.1%, and the results are presented in Figure 4a. During the titration with AG 0.1%, the new broad emission maximum at 514 nm appeared after the addition of only 10 μL of AG, suggesting a strong interaction with Dox. Its intensity gradually increased until the addition of 50 μL of AG, and then stabilized even at very large amounts of AG (100 μL). At higher concentrations of AG, one can observe a slight increase in intensity and a redshift by 7 nm of the maximum at 555 nm (Figure 4a). The interaction of AG with Dox is also supported by the variation of the ratio between the fluorescence intensity at λ_em_ = 555 nm and λ_em_ = 596 nm with the concentration of AG. The ratio between the fluorescence intensity gradually increased from 0.54 (Dox initial) to 0.66 after the addition of 50 μL of AG, and then slightly decreased to 0.64.

Titration of Dox with R-SH 0.1% led to a slight increase of emission at 596 nm and 636 nm at λ_exc_ = 470 nm (Figure 4b). The changes in the emission at 636 nm are due to a more hydrophobic environment around the Dox molecule induced by the presence of R-SH. By increasing the concentrations of R-SH, the ratio between the emission bands at 555 mm and 596 nm decreased from 0.54 (Dox) to 0.46 (50 μL R-SH).

Based on titration data, the binding constant was estimated by the Benesi–Hildebrand Equation (7):(7)$$\frac{F_{1}-F_{0}}{F-F_{0}}=1+\frac{1}{K_{b}\times \left[L\right]}$$
where *L* is AG or R-SH; *F*_0_ is the fluorescence intensity in the absence of *L*; *F*_1_ is the fluorescence intensity after the addition of the maximum [*L*]; *F* is the fluorescence intensity at each added [*L*]; and *K_b_* is the binding constant.

The titration data showed a good linearity (R^2^ = 0.996) (Appendix A) confirming a 1:1 association between Dox and *L*. The corresponding binding constant for Dox: AG is *K_b_* = 9.75 × 10^4^ M^−1^, while for Dox: R-SH is 5.72 × 10^4^ M^−1^, proving a strong interaction of Dox with both AG and R-SH ligands. The standard Gibbs free energy was determined according to Equation (8):ΔG^0^_binding_ = −2.303 *RT* log*K_b_*(8)
where *R* is the universal gas constant (8.3145 J mol^−1^ K^−1^); T is temperature (Kelvin); and *K_b_* is the association constant from the Benesi–Hildebrand Equation (7). The calculated value of ΔG^0^_binding_ is −28.48 kJ mol^−1^ (−6.807 kcal mol^−1^) for Dox: AG and −27.16 kJ mol^−1^ (−6.49 kcal mol^−1^) for Dox: R-SH, characteristic of H-bonding interactions [48]. This result shows that, in aqueous dilute solutions, both functional groups are able to bond to Dox with similar strength.

### 3.5. IR Spectroscopy

The IR spectra of the silica before and after the encapsulation of Dox (denoted by the letter “D”) are presented in Figure 5, with emphasis on the regions of interest. Besides the silica framework pattern and the bands due to the Si-OH groups and C-H bonds stretches, some bands due to CTAB appear in Sil2 and Sil3 (for example, that at1487 cm^−1^ specific for asymmetric CH_3_ deformation vibration in N-(CH_3_)_3_ groups) (Figure 5). The characteristic stretching vibrations of Dox assigned to NH_2_, C=O and C=C/C-N bonds were better highlighted by the second derivative in the spectral regions 3400–2800 cm^−1^ (3240 cm^−1^ in MSil2D, 3300 cm^−1^ in MSil3D and 3383/3324 cm^−1^ in MSil1D)) and 1780–1560 cm^−1^ (1724 cm^−1^ and 1582 cm^−1^ in Sil2D, 1732 cm^−1^ and 1582 cm^−1^ in Sil3D, 1726 cm^−1^ and 1582 cm^−1^ in MSil2D and MSil1D, 1726 cm^−1^ and 1580 cm^−1^ in MSil3D (Figure 5, inset graphs).

The interaction of Dox with the thiol and glucose functional groups of silica was also observed by the second derivative of the spectra in the 800–400 cm^−1^ spectral range (Figure 6). The weak differences between Sil2 and Sil2D are due to the presence of CTAB on the surface of the silica, supported by the band at 724 cm^−1^. The band at 687 cm^−1^ attributed to C=C ring bend in Dox confirmed the encapsulation of Dox in Sil2. In the Sil3 spectrum, the bands at 716 and 698, 570 and 530 cm^−1^ are specific for C-O-C bending vibration in glycoside and C-S-C bonds, respectively. The interaction of Sil3 with Dox led to the blueshift of maxima at 716 cm^−1^ and 570 cm^−1^ by 16–20 cm^−1^ in Sil3D (Figure 6).

The presence of Dox in MSil2D and MSil3D was confirmed by the absorption maxima at 805 cm^−1^ specific for the C=H bend in Dox overlapped with the Si-O bending (Figure 7). The bands at 742 cm^−1^ in MSil2D and 754 cm^−1^ in MSil3D are assigned to thiol and glucose groups, respectively, associated with Dox, while the bands at 700/702, 600/598 cm^−1^ and 514 cm^−1^ in MSil2D and MSil3D are due to the C=C ring bend, redshifted by 13–15 cm^−1^, and C-H out-of-plane in aromatic ring, respectively, also redshifted by 10–22 cm^−1^ compared to the initial Dox (687 cm^−1^/577 cm^−1^/504 cm^−1^). These higher wavenumber shifts also supported the Dox–Dox hydrogen bonding interactions. In all of the silica samples, the absorption maxima at 468 and 400 cm^−1^ are specific for Si-O rocking and in-plane bending vibrations, respectively.

### 3.6. Encapsulation of Dox

The encapsulation of Dox was monitored by the spectrophotometric analysis of the supernatant after the dispersion of silica in Dox solution 0.1% in PBS pH 7.4.The maximum encapsulation efficiency and loading capacity presented in Table 3 were obtained after 24 h encapsulation, except for samples MSil2 and MSil3, which reached their maximum performance after repeated encapsulation from a more concentrated Dox solution (~0.2%).

High encapsulation efficiency and loading capacity (more than 75 μg/mg of silica) were obtained in the tested conditions after 24 h. The mesoporous silica MSil2 and MSil3, which encapsulated practically the entire amount of Dox in the first batch, were used in a second loading cycle, when maximum LC values of 225 μg/mg for MSil2 and 174 μg/mg for MSil3 were reached after another 24 h interval. When comparing these two samples, the one modified with glucose had slightly lower loading parameters, probably due to the bulkiness of the glucose moiety, which would restrict the penetration of Dox into mesopores. Overall, the silica loading efficiency recorded for all of the tested materials exceeds those reported in the literature for silica nanotubes [49], and is similar as reported for MCM-41 silica [50] and for magnetic mesoporous NPs [16].

Besides the physical interactions discussed above, other features of the silica materials might seriously influence the encapsulation behavior, such asthe mesopores’ size, the surface area and the hydrophilicity. Indeed, the mesoporous silicas MSil2 and MSil3 are more hydrophilic than MSil1, while their pores are free, as opposed to their precursors, which contain CTAB surfactant.

The Zeta potential values of the initial and Dox-loaded silicas are presented in Table 4. The surface charge is an important parameter, e.g., significantly influencing the excretion of mesoporous silica nanoparticles [10]. The nonporous silica Sil2 has a positive Zeta potential, due to CTAB. The Zeta potential of MSil2 after surfactant removal is negative, but to a lower extent than previously reported for all-thiol-organosilica [51], in agreement with the lower content of SH groups in our case. The methyl-modified mesoporous silica MSil1 has a positive charge, which was postulated to allow binding to DNA and proteins [52], and considered beneficial in improving oral absorption. If neat Sil2 and MSil2 are compared with their glucose-modified homologues, the changes are spectacularly different:the decrease of Zeta potential is much more pronounced in Sil3, indicating a substantial modification of the surface, while little difference in absolute value is registered in the case of the mesoporous pair of silica, due to a more important modification at the pores’ level. As a general observation, when Dox was loaded, the Zeta potential values decreased more or less, depending on the main localization of the drug molecules (at the surface or in the pores). In particular, a greater change in Zeta potential was registered in Dox-loaded MSil3 than in the case of MSil2, which could indicate that the pores of MSil3 may be partially blocked by the glucose modification, leading to a higher amount of Dox bonded to the surface, in line with the lower loading capacity observed for MSil3 compared with MSil2.

### 3.7. Release of Dox in Different PBS

Dox release behavior was assessed spectrophotometrically by immersing the loaded silica in buffer solution, pH 7.4, 5 or 2.6. The maximum release values, calculated as % of Dox released after 72h from the amount of drug encapsulated (Equation (3)), are presented in Table 5, as well as the amounts of the drug released reported to the mass of silica (Equation (4)). We have to mention that the values for release at pH 5 in Table 5 are reported to the maximum amount of the drug encapsulated from a more concentrated solution, and, in the case of MSil2 and MSil3, to double encapsulation. In addition, in each release experiment, the extinction coefficient of Dox measured at the corresponding pH was used.

It was observed that both encapsulation efficiency and cumulative release were comparable with the literature data for silica nanoparticles containing pH triggered nano-valves. The release at pH 5 from samples loaded in dilute and concentrated solutions, respectively, (maximum loading) is compared in Appendix A. Similar amounts of Dox were released from the single-loaded samples, irrespective of the initial concentration, while notably high amounts of the drug were released from the mesoporous silicas MSil2 and MSil3 at their highest loading. In terms of the Dox amount released per mass of silica, the silica without mesopores having glucose groups (Sil3) was able to release almost twice the amount of Dox released by Sil2, without glucose, at pH 5. The mesoporous silicas released more Dox than the nonporous ones (Table 5) at pH 5 (10-fold more in the case of MSil2) and at pH 7.4, based on the higher amount encapsulated. A significant difference between MSil2 and the glucose-modified homologue can be observed, depending on the pH: higher release for MSil3 at pH 2.6 and 7.4, but lower at pH 5 (Table 5). At pH 2.6, the nonporous silicas, containing CTAB, exhibited higher release capacity than the mesoporous ones.

All ofthese differences have to be correlated with the complex panel of intermolecular, pH-dependent physical interactions. For example, the electrostatic interactions between ionized silanol and protonated Dox are more pronounced above pH 4.5 (see Table 2), stabilizing the drug within the silica framework, while at pH 2.6, all of the silanol groups are protonated, thus the level of interaction is lower, and this effect might be enhanced by the release of CTAB in acidic pH, uncovering the silica surface.

The extracellular space of the tumor is characterized by low oxygen and higher acidity, so that a higher release of Dox at pH 5 would improve the therapeutic efficiency, by increasing the Dox internalization into cells by endocytosis [53].

The release profile (Figure 8), expressed as cumulative release % from the encapsulated drug in dilute solution, was similar for most of the silica materials, reaching a plateau after ca. 4h, with two notable exceptions: sample MSil1 which had a sustained release during 24 h in pH 7.4 (Figure 8a); and sample MSil2 showing sustained release for 24 h at pH 5 (Figure 8b). The cumulative release was about 25–75% from the encapsulated amount in pH 7.4, 54–98% in pH 5, and 29–60% in pH 2.6 at 37 °C (Figure 8c). The release profiles at pH 5 for the samples loaded from the concentrated solution (Appendix A) showed the same sustained release trend for MSil2.

Two mathematical models were used to analyze the drug release kinetics: Higuchi (Equation (5)) [32] and the Korsmeyer–Peppas equation (Equation (6)) [33]. Both models can be applied when M_t_/M_∞_ ≤ 60%, as is generally the case here. By fitting the data on the increasing regions of the release profile, the coefficients summarized in Table 6, Table 7 and Table 8 were obtained.

In the case of the release at pH = 7.4, the correlation coefficient for the Korsmeyer–Peppas equation (which applies for porous hydrophilic materials) was slightly higher (Table 6). When M_t_ vs. t^1/2^ graphs were analyzed (Higuchi equation) an inflexion point was observed in all of the cases at 90 min, which would indicate two successive processes. Taking these data separately, very high correlation coefficients for the Higuchi equation were found, showing that the release process can be described by two diffusion steps. In all of the cases, the release profile was similar, with a maximum cumulative release at around 4.5 h, except for MSil1, which showed a sustained release for 24 h (Figure 8a).

At pH 5, samples Sil2, Sil3 and MSil2 had a sustained release practically forthe entire observation period (72 h), but Sil2 and Sil3 presented a very low release rate after the first 4h (the sustained release after this time was ignored in the kinetics evaluation), while sample MSil2 showed a pronounced continuous release within the first 24 h with a cumulative release of 98.25% (Figure 8b). In Table 7, the release kinetic parameters were calculated for the first 3.5 or 4.5 h, except for MSil2, where the kinetic data are given for 24 h. The Higuchi model described better the release behavior when applied for two release processes for all of the samples, except Sil2, where a single process was observed. The Korsmeyer–Peppas model was generally applied for the same time frame as for Higuchi, except in the case of MSil1, where a release burst was observed, and reaching 97% within 3.5h. The release kinetics of the samples loaded from more concentrated solutions are summarized in Appendix A.

Comparing the release profiles forcancer tissue-associated pH and physiological conditions, and taking as a criterion the generally accepted requirement for sustained release in cancer cells and relatively low release levels in the healthy tissue, it appears that the best candidate would be MSil2 with the thiol groups.

In pH = 2.6, the Higuchi model best described the release from the silicas without mesopores. For the mesoporous silica, the Korsmeyer–Peppas equation fitted better, with release exponents (generally below 0.5) indicating a preponderant diffusion process which obeys Fick’s law (Table 8). An inflexion point in the M_t_ vs. t^1/2^ representations was observed only for the mesoporous silica, indicating two diffusion steps, a diffusion from the surface in the first 90 min (or even 50 min in the case of MSil1), and a subsequent diffusion from the mesopores within the next 2 or 2.5 h, respectively. Interpreting these data in terms of possible applications e.g., in oral formulations, it appears that the high release of Sil3, Sil2 and MSil3 in the first hours would be useful for stomach cancer, while the low release at pH 2.6, correlated with high and sustained release at pH 7.4 in the case of MSil1 would be interesting for other regions of the gastrointestinal tract [54]. In the latter case, the positive Zeta potential and high hydrophobicity might be favorable for oral administration.

As can be seen in Table 6, Table 7 and Table 8, the *p*-value was 4.24 × 10^−6^ (pH 2.6), 3.14 × 10^−12^ (pH 5) and 6.14 × 10^−13^ (pH 7.4), being much smaller than the significance level of 0.05, proving that there is a statistical difference between the Dox release of tested silica materials in different pH media.

### 3.8. In Vitro Cytotoxicity

The cell viability was evaluated by CellTiter-Glo^®^ assay and the effect of Dox-loaded silica samples was compared with that of “free” Dox solutions. For these experiments, freshly prepared samples were used, wherein Dox was encapsulated with a slightly different procedure that is rapid encapsulation from the 0.1% solution, under sonication, and the encapsulated amounts were determined spectrophotometrically. In order to better compare the samples, in this paragraph the Dox-loaded silica are depicted with the letter “D”. The concentration of Dox contained in each sample is shown above the graphs of cell viability vs. sample concentration in Figure 9, and it is clear that a very low amount of Dox is actually present in the samples at the tested concentrations (maximum 2 μg/mL).

First, we have to observe that Dox itself had a different effect on different cell lines, the lowest toxicity being registered for HeLa cells at low Dox concentrations, while at the higher Dox concentration of 50 μg/mL, the cell viability was 30–40%. This apparently weak effect can be ascribed to partial degradation of Dox in the cell culture medium within the long observation time [55]. On the other hand, all ofthe Dox-loaded silica samples showed a stronger cytotoxic effect than the free Dox solution, with lower cell viability. It appears that at Dox concentrations as low as 1–2 μg/mL in silica samples, the effect was similar or even stronger than for the free drug at 50 μg/mL. Thus, other factors with synergistic effects on cell viability should be considered. The insignificant effect of neat silica was previously reported by MTT tests, while the thiol groups significantly reduced the toxic effect of gold nanorods on human H460 cells, compared with CTAB-coated particles [56]. Here, to assess the role of unloaded silica, a cellular viability assay on NDHF (fibroblasts) and in cancer cell lines (MeWo and HeLa) in the presence of mesoporous silica materials was performed (Figure 10).

All of the mesoporous silica revealed a good biocompatibility with normal cells (NHDF), even at a concentration of 50 μg/mL. A slight decrease in viability on HeLa cell lines at a lower concentration (1.56 μg/mL) was exerted by MSil1 (90% cell viability), while for MSil2 and MSil3 the same effect was observed at a concentration of 12.5 μg/mL. At the same concentrations, silica exerted less toxicity on MeWo cells. A relatively increased inhibitory activity on the HeLa cell line was exerted at a higher concentration, 50 μg/mL, by all of the silica materials, the cell viability decreasing by 20–25% (Figure 10). A decrease in cell viability by 20% for 120 nm MSNs was reported on MCF-7 and MCF-7/ADR cell lines at a concentration of 10 μg/mL [14].

When the effect on all of the three cell lines of 50 μg/mL of mesoporous silica, containing ca. 2 μg/mL encapsulated Dox, is compared with the corresponding concentration of free Dox and pristine silica, a synergistic effect is obvious. In Appendix A we plotted the difference in cell viability of the Dox-free and Dox-loaded mesoporous silica, as a measure of their net cytotoxic effect. It results that the most pronounced synergistic action was provided by MSil2, followed by MSil1 and MSil3. The observed behavior is probably due to the increased in vitro stability of the encapsulated Dox compared with the free drug, correlated with the gradual release from the pores. It was reported by cellular uptake and in vivo biodistribution experiments [14] that Dox-loaded unmodified MSNs induced a higher accumulation of DOX in drug resistant tumors than free DOX, and decreased accumulation in other tissues. A similar effect was reported for Dox-loaded MSNs with a particle size of 120 nm on MCF-7 and MCF-7/ADR cell lines, where the silica particles have proven to work as a carrier with a high potential to enhance the cytotoxicity [14].

The glucose moieties in the structure of silica did not significantly influence the cell viability in the absence of Dox (Figure 10), but increased the cell viability in the presence of Dox in all of the cases compared to the corresponding precursors without glucose (Figure 9). Comparing Dox-loaded Sil2 with Sil3 and MSil2 with MSil3, it can be observed that the presence of glucose especially enhanced cancer cells’ viability; glucose tends to nourish the cancer cells more than the fibroblasts. On the other hand, our experiments were completed in standard cell culture medium, containing 1g/L of glucose. It was demonstrated that sugar modification of PAMAM-Dox conjugates, besides inducing pH-dependent drug release, strongly increased the cytotoxicity in MCF-7 cells cultured in the medium without glucose [25]. The behavior was ascribed to the “Warburg effect” [57], which is a form of modified cellular metabolism found in cancer cells, associated with high glycolysis rates. Assuming this hypothesis, the glucose-modified silica materials could play the role of a trap (“Trojan horse”) for starved cancer cells, but this remains to be studied in the future. In the meanwhile, the obtained results might be regarded in relationship with the physical interactions between Dox and glucose, and, consequently, with the release and stability of Dox. Since glucose bonded on silica did not influence the cell viability, we suppose that the increased stability of Dox in the presence of glucose moieties and possibly specific interactions with cancer cells would be responsible for the observed behavior.

For nonporous silica, the presence of CTAB enhanced the toxicity, as was previously reported [58]. CTAB is known for its apoptosis-promoting role [59] and was found to synergistically increase the cytotoxicity of Dox-loaded MSNs against MCF-7, greatly improving the intracellular accessibility of the poorly water-soluble drugs [60].

Selectivity for HeLa cells compared to fibroblasts and MeWo cells was practically observed for all of the samples, at a different concentration range, and it was more pronounced in the nonporous silica materials. The observed selectivity for cervical cancer cells may be very interesting for intravaginal administration, especially for the samples which presented a better release capacity in pH 7.4, based on an increased pH in abnormal cervical cytological smears (from around 4 up to 10) [61]. The mesoporous silicas also presented some selectivity for the MeWo cells compared with NHDF at relatively high concentration, and a good cytotoxic effect, which might be promising for topical applications in skin cancer.

## 4. Conclusions

Mesoporous and nonporous functionalized (methyl, thiol and glucose groups) silica NPs were synthesized and loaded with doxorubicin hydrochloride in order to investigate their specific interaction. The chemical modification of the thiol-functional silica with a double bond glucoside was completed for the first time, by green thiol-ene photoaddition. These silica materials were investigated for encapsulation and release of Dox in different pH, as well as for the cytotoxicity on three cell lines. High loading efficiency values were found for mesoporous silica, especially for thiol- and glucose-modified ones. The release capacity was also high, reaching 98% and showing sustained release in certain cases. A strong cytotoxic effect was observed in all of the loaded silica at equivalent Dox concentration of around 2 μg/mL, much more pronounced than that of free Dox, and non-loaded silica. Selectivity for HeLa cells compared to fibroblasts and MeWo cells was observed for all of the samples, at a different concentration range, which may be of interest for the treatment of cervical cancer and intravaginal administration. The mesoporous silicas also presented some selectivity for MeWo cells compared with NHDF at a relatively high concentration, which, cumulated with the good cytotoxic effect, may be promising for topical applications in skin cancer.

## Data Availability

The data presented in this study are available on request from the corresponding author.

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
