# Peer review of "Functionalized Mesoporous Silica as Doxorubicin Carriers and Cytotoxicity Boosters"

_nanomaterials, 2022, doi:10.3390/nano12111823_

Round 1

Reviewer 1 Report

In the manuscript entitled “Thiol- and glucose-functionalized mesoporous silica as doxorubicin cytotoxicity boosters” the authors report on the synthesis of mesoporous silica nanoparticles functionalized with thiol and glucose groups and on their use as Doxorubicin carrier for cancer treatment.

The topic is very relevant and supported by appropriate and recent literature references, and the experimental data have been reported clearly.

My only perplexity concerns the mismatch between what is stated in the title of the paper and the results obtained from the cell viability tests.

By reading the title of this paper, I expected to find that the thiol- and glucose-functionalized mesoporous silica nanoparticles loaded with doxorubicin were the most effective in inducing cancer cells death. But if we compare the results of the cell viability tests obtained by silica nanoparticles loaded with the same amount of doxorubicin what emerges is that:

  • The enhanced cytotoxic effect of porous and non-porous silica nanoparticles loaded with doxorubicin with respect to free doxorubicin does not depend from the functionalization with thiol or glucose groups, with comparable results obtained also in the case of MSil1D.
  • MSil1D is as cytotoxic as MSil2D and more cytotoxic than MSil3D.
  • The cytotoxic effect of non-porous silica nanoparticles (Sil2D and Sil3D) is more pronounced with respect to the corresponding porous nanoparticles and selective only in the case of HeLa cells.

Overall, these data denoted that the functionalization of the mesoporous silica nanoparticles does not effectively booster the cytotoxicity of doxorubicin.

My suggestion is to elaborate a more appropriate title for the paper and to adequately discuss the results of the cell viability tests in the conclusion section.

For this reason I recommend the publication of the paper in Nanomaterials after major revisions.

Author Response

Please see in the attachment.

Reviewer 2 Report

Although this is a sound piece of work, some amedments must be done before publishing. The main one is related with the conclusions. It seems authors wrote a summary rather than a conclusions section.

Lines 750-753 :The mesoporous silica materials were characterized in terms structure (FT-IR, Raman), morphology (TEM), porosity (nitrogen sorption –desorption) and Zeta potential. The drug encapsulation by physical interactions was highlighted by UV-, zeta potential, FT-IR, fluorescence, and molecular dynamics investigations.  should be eliminated

Lines 765-766: "while the loaded  silicas exhibited a synergistic effect of silica and Dox in all tested cell lines, probably due  to better stability and slow release of Dox from the mesopores.      is rather a discussion point and should be removed from this section

 And in general the section is just a summary of results or extended abstract and a whole re-writing is suggested..

Experimental section. It would be easier for the reader  if the description of techniques follow the same order than the results section.

Resuts

Table 1. Whether the whole sorption-desorption curve has been completed, It would also be interestesting to show and study the shape of the pores in terms of potential chemical modification and drug loading.

Figure 1. please correct the baseline of he glucoside modified spectrum (MSil3, red line). and since you are marking tha band at 2576 cm -1 in the figure, please also indicate the assignation as well ( not only in the main test )

TGA analysis, why the glucoside derivative linked.. is only study in terms of the  percentage of weight residue? There are differences in the steps that can be significant as well.

Line 381. hydrophobic sample was MSil1, while the most hydrophilic one was MSil3, containing glucose,.    Is there wetting results that deserve mentioning? or is only the inmediiate soaking ability displayed in figure S2? In the last case , the sentence must be softened.

line 502 observed by second derivative of the spectra in the 800-400 cm
1 spectral range (Figure 503 S5 and S6). I suggest to incorporate these spectra into the main body of the manuscript since they are the important analysis made and move the whole spectra into the supplementary material

Figure 6. Please add statistics of the results. Also in the corresponding tables

Line 695 The cytotoxicity of the silica depends on the concentration, the size, morphology and the silica functionality.... This sentence is only valid whether significant diferences on the analysis are shown

Round 2

Reviewer 1 Report

I want to  thank the authors for taking into consideration the reviewer suggestions. The revised paper has been significantly improved both in the results and discussion  and in the conclusions sections. For this reason I accept the paper as is for the publication in Nanomaterials 

Author Response

Thanks Reviewer 1 for the comments!